# Implications of the Niche Partitioning and Coexistence of Two Resident Parasitoids for *Drosophila suzukii* Management in Non-Crop Areas

**DOI:** 10.3390/insects14030222

**Published:** 2023-02-23

**Authors:** María Josefina Buonocore Biancheri, Segundo Ricardo Núñez-Campero, Lorena Suárez, Marcos Darío Ponssa, Daniel Santiago Kirschbaum, Flávio Roberto Mello Garcia, Sergio Marcelo Ovruski

**Affiliations:** 1Planta Piloto de Procesos Industriales Microbiológicos y Biotecnología (PROIMI-CONICET), División Control Biológico, Avda. Belgrano y Pje. Caseros, San Miguel de Tucumán 4000, Tucumán, Argentina; 2Centro Regional de Investigaciones Científicas y Transferencia Tecnológica de La Rioja (CRILAR), Provincia de La Rioja, UNLaR, SEGEMAR, UNCa, CONICET, Entre Ríos y Mendoza s/n, Anillaco 5301, La Rioja, Argentina; 3Departamento de Ciencias Exactas, Físicas y Naturales, Instituto de Biología de la Conservación y Paleobiología, Universidad Nacional de La Rioja (UNLaR), Av. Luis M. de la Fuente s/n., Ciudad de La Rioja 5300, La Rioja, Argentina; 4Dirección de Sanidad Vegetal, Animal y Alimentos de San Juan (DSVAA)—Gobierno de la Provincia de San Juan, CONICET, Nazario Benavides 8000 Oeste, Rivadavia 5413, San Juan, Argentina; 5CCT CONICET San Juan, Argentina Av. Libertador Gral. San Martín 1109, Capital 5400, San Juan, Argentina; 6INTA Estación Experimental Agropecuaria Famaillá, Tucumán Ruta Prov. 301, km 32, Famaillá 4132, Tucumán, Argentina; 7Cátedra Horticultura, Facultad de Agronomía y Zootecnia, Universidad Nacional de Tucumán, San Miguel de Tucumán 4000, Tucumán, Argentina; 8Departamento de Ecologia, Zoologia e Genética, Instituto de Biologia, Universidade Federal de Pelotas, Pelotas 96000, Rio Grande do Sul, Brazil

**Keywords:** spotted-wing drosophila, drosophilid abundance, pupal parasitoid coexistence, ecological profiles, feral fruit host, non-crop environment

## Abstract

**Simple Summary:**

*Drosophila suzukii*, internationally known as the spotted-wing drosophila (SWD), is an invasive insect pest that mainly causes economic damage to fresh and healthy, as well as soft and stone, fruit crops. The SWD has quickly spread throughout all the Argentinean fruit-growing regions. Natural enemies, such as parasitoids, can be an important environmental friendly tool within an SWD management strategy. However, understanding the biological mechanisms that enable the coexistence of different parasitoid species in a particular environment is essential to improve their use as biocontrol agents. Therefore, this study assessed the coexistence of two resident pupal parasitoids, *Trichopria anastrephae* (*Ta*) and *Pachycrepoideus vindemiae* (*Pv*), on SWD-infested guava and peach in non-crop areas of northwestern Argentina, based on spatial (microhabitat) and/or resource (host flies) differentiation. Results revealed that both biological mechanisms might mediate the coexistence of these two pupal parasitoid species. *Ta* showed a preference for resident saprophytic drosophilid puparia located mainly inside fruit flesh, whereas *Pv* searched for the host in less competitive habitats, such as in the soil or outside fruit flesh, where SWD puparia prevailed. Such a differential exploitation of host microhabitats influenced parasitoid efficiency in suppressing SWD populations. The combined use of both parasitoid species may be advisable for local SWD management.

**Abstract:**

Understanding the mechanisms associated with the coexistence of competing parasitoid species is critical in approaching any biological control strategy against the globally invasive pest spotted-wing drosophila (=SWD), *Drosophila suzukii* (Matsumura). This study assessed the coexistence of two resident pupal parasitoids, *Trichopria anastrephae* Lima and *Pachycrepoideus vindemiae* Rondani, in SWD-infested fruit, in disturbed wild vegetation areas of Tucumán, northwestern Argentina, based on niche segregation. Drosophilid puparia were collected between December/2016 and April/2017 from three different pupation microhabitats in fallen feral peach and guava. These microhabitats were “inside flesh (mesocarp)”, “outside flesh”, but associated with the fruit, and “soil”, i.e., puparia buried close to fruit. Saprophytic drosophilid puparia (=SD) belonging to the *Drosophila melanogaster* group and SWD were found in all tested microhabitats. SD predominated in both inside and outside flesh, whereas SWD in soil. Both parasitoids attacked SWD puparia. However, *T. anastrephae* emerged mainly from SD puparia primarily in the inside flesh, whereas *P. vindemiae* mostly foraged SWD puparia in less competitive microhabitats, such as in the soil or outside the flesh. Divergence in host choice and spatial patterns of same-resource preferences between both parasitoids may mediate their coexistence in non-crop environments. Given this scenario, both parasitoids have potential as SWD biocontrol agents.

## 1. Introduction

In nature, resident and introduced parasitoid species may be able to coexist in the same host species by niche partitioning, i.e., the process by which competing species move into different patterns of resource use or different niches [1], or through different ecological profiles and life histories [2,3,4]. Among various mechanisms enabling the coexistence of competing parasitoids, the temporal and spatial partitioning of resources may be highlighted [5]. As a result, the co-occurrence of competing parasitoid species may depend on the occupation of competitor-free spaces [6]). Thus, niche differences may imply divergence and a competitive avoidance history [7]. Consequently, any information related to the mechanisms associated with the coexistence of competing parasitoid species is essential in addressing any biological control strategy against invasive insect pests [8,9].

The globally invasive pest spotted-wing drosophila (=SWD), *Drosophila suzukii* (Matsumura) (Diptera: Drosophilidae), a native of South East Asia [10] and currently occurring on all continents [11], has quickly spread throughout all the fruit-growing regions of Argentina since it was first recorded in 2014 [12]. The SWD is an economically important pest of small, soft, and stone fruits worldwide, because females lay eggs in fresh, healthy, ripening fruit [13]. Curiously, SWD is one of the few *Drosophila* species that has evolved into a serrated ovipositor, which allows females to drill into the skin of healthy fruits to oviposit inside the fleshy mesocarp [14].

Fly larvae feed deep into the fruit’s fleshy mesocarp, resulting in fruit rot. Mature larvae emerge from the fruit to pupate mainly in the soil, although larvae usually also pupate inside fallen fruit or beneath the fruit without burying themselves, remaining attached to the fruit skin [15]. Although SWD is mainly a pest of berry and cherry crops, this dipteran is highly polyphagous, as it has a broad host fruit range, mainly throughout Asia, Europe, and America [10]. In addition to crop host species, mainly Rosaceae, the SWD larva can develop in both native and exotic fruit of ornamental and wild non-crop hosts [15].

The SWD is found in 64 host plants in 25 families in Latin America. Although most hosts are exotic in this region, about 39% are native plants that can become alternative hosts and reservoirs of the pest in the intercrop period [11]. In Argentina, 15 fruit species have been recorded as hosts of SWD, including both feral guava (*Psidium guajava* L.) (Myrtaceae) and feral peach [*Prunus persica* (L.) Stokes] (Rosaceae) [11,16]. These SWD host plants are among the most common and widespread exotic feral fruit species growing in wild vegetation patches, adjacent to commercial fruit crops in northwestern Argentina. Natural infestation levels by SWD in feral guava and feral peach range between 5 and 10% per kg of sampled fruit [12,16]. Guava is not commercially grown in Argentina. It can only be found as an ornamental plant in gardens or as a backyard fruit tree or scattered in wilderness areas with high levels of human disturbance. Peach is cultivated in northwestern Argentina at a very low scale, with no influence on the local or national supply. Peaches are mainly grown in the central–western and south–northeastern regions of Argentina. However, cultivated peaches were not reported to be infested by the SWD, although there are some records of SWD adults caught in liquid traps placed inside the commercial peach orchard (SWD). In spite of the SWD infesting cultivated peaches in some Asian, European and North American countries, in fruits mainly with previous wounds, it is not a natural host of this pest [15].

As SWD spreads almost worldwide, many countries have immediately adopted preventive and intervention measures to minimize economic losses. Thus, SWD mitigation strategies, including exclusion netting, mass trapping supported on attractant-based traps, crop sanitation, and chemical and biological controls, were implemented [13], while the sterile insect technique is currently being evaluated [17]. Concerning biological control, natural enemies may be particularly important as an eco-friendly tool in a network of SWD management strategies, maximizing ecosystem services’ benefits. In this regard, information on wild host fruit status, on which SWD populations may increase, is critical to support management strategies, particularly in wilderness environments surrounding commercial fruit crops [18,19,20]. Therefore, it is imperative to understand better the trophic interactions between SWD and the components of newly invaded landscapes regarding the available hosts, other frugivorous dipterans, and natural enemies [15]. Among SWD’s biological controllers, parasitoid hymenopterans are the best studied and most likely to be successful [21,22,23]. An assemblage of resident koinobiont larval and idiobiont pupal parasitoids has been associated with SWD in crop and non-crop areas of northwestern Argentina [16]. Among all these species, two pupal parasitoids, *Trichopria anastrephae* Lima (Hymenoptera: Diapriidae) and *Pachycrepoideus vindemiae* Rondani (Hymenoptera: Ptromalidae) are commonly abundant, and are often found foraging in search of host puparia on the same fallen fruit [16]. The diapriid *T. anastrephae*, native to South America [24], is a pupal endoparasitoid whose female lays the egg into the hemocoel of the host fly pupa [25]. In contrast, the cosmopolitan *P. vindemiae* not only attacks a wide range of dipteran species but is also a pupal ectoparasitoid, because the female lays the egg inside the space between the puparium shell and host pupal body [26]. Both parasitoids were recorded from tephritid puparia, particularly from *Anastrepha* spp. and *C. capitata*, in Argentina and Brazil [27]. Furthermore, *P. vindemiae* was associated with *D. suzukii* in berry and cherry crops of different Argentinian regions [28]. Although both are idiobiont pupal parasitoids, *T. anastrephae* is an endo-parasitoid, and *P. vindemiae* is an ecto-parasitoid, so they belong to different guilds [29]. A parasitoid guild can be acceptably defined as two or more sympatric species that equally exploit a particular developmental stage of the host [30].

Both *P. vindemiae* [31,32] and *T. anastrephae* [33,34] can be successfully lab-reared on SWD puparia and have shown high potentials as *D. suzukii* biocontrol agents [21,22]. However, competitive tests, undertaken under lab conditions, between *P. vindemiae* and *T. anastrephae* [25], and also with the cosmopolitan *Trichopria drosophilae* Perkins [31], showed that both diaprid species out-competed the pteromalid. The studies above revealed the superiority of the two *Trichopria* species over *P. vindemiae* in intrinsic competitiveness and foraging efficiency. In addition, SWD’s resident parasitoid surveys in Brazil recorded a predominance of *T. anastrephae* on *P. vindemiae* [35]. In contrast, two interesting shreds of evidence have been revealed in a recent survey carried out at a non-crop habitat overgrown by feral peach trees in Tucumán, northwestern Argentina [16]: (1) *P. vindemiae* mostly parasitized SWD puparia, and (2) *Trichopria* sp., identified later as *T. anastrephae*, predominantly parasitized puparia from drosophilids of the *Drosophila melanogaster* group. In light of the preceding information, a question arises: how do these competing pupal parasitoids coexist, attacking both SWD and saprophytic *Drosophila* spp. Puparia, in the same fruit at the same time? Therefore, it was hypothesized that the coexistence of both pupal parasitoids on drosophilid-infested fruits in wild vegetation areas of Tucumán results from niche segregation, including spatial partitioning or resource partitioning, or both. In the first scenario, it is assumed that *P. vindemiae* occupies the *T. anastrephae*-free space provided by SWD puparia from microhabitats poorly exploited by the diaprid species. For the second option, it is postulated that in a shared niche situation, *P. vindemiae* is more specialized to inhabit the newly introduced host species, taking into account its cosmopolitan status and because it is far more generalist than *T. anastrephae*. A third situation involves a combination of both parasitoid species. Based on these assumptions, it is predicted that: (1) the distribution of drosophilid puparia in several microhabitats associated with the fruit will reveal differences in the space-use pattern between the SWD and resident drosophilid species; (2) *P. vindemiae* females will focus their search for, and attack, SWD puparia located outside the internal part of the fallen fruit; (3) parasitism on SWD puparia by *P. vindemiae* will increase when the density of resident drosophilid puparia highly exceeds that of SWD; (4) should such a spatial or a resource partitioning occur, *P. vindemiae* females will reduce their competitive interactions with *T. anastrephae* and thus avoid endangering their offspring. To test these predictions, a survey of drosophilid puparia scattered in different microhabitats associated with the fruit, e.g., inside the mesocarp, outside it but attached to the fruit, and buried beneath the fruit of non-crop hosts, such as feral guava and peach, in two disturbed wild vegetation sites of Tucumán was performed. Microhabitat differentiation was addressed for host puparia sampling based on the strong influence of the microhabitat type, e.g., soil or canopy, on the parasitoid assemblage, associated with saprophytic drosophilids consuming decaying organic matter, rather than habitat type [36].

The findings of this study will be useful for planning SWD biological control strategies within an area-wide integrated pest management approach [21] in Argentinian fruit-growing regions and elsewhere around the Americas.

## 2. Materials and Methods

### 2.1. Study Area

The study was carried out in an area characterized by a mosaic of environments, such as suburban sectors occupied by housing within a secondary rainforest matrix, with a predominance of exotic plants, citrus crops, and mountain slopes, slightly disturbed with a high presence of indigenous plant species. This area, located in Horco Molle, Yerba Buena district, Tucumán province, northwestern Argentina, belongs originally to the first vegetation level of the Yungas rainforest eco-region, called “Premontane Forest” [37]. The Yungas is a narrow strip of South American subtropical montane rainforest located along the eastern slopes of the Andes mountain range, starting from Peru and extending into northwestern Argentina [38]. The study area belongs to the southernmost extension of the Yungas. Two sampling sites were chosen within the study area (Figure 1). Site #1, located at 26°48′ S latitude and 65°19′ W longitude, and 520 m altitude, was a 2 ha patch of secondary structure rainforest with feral guava trees predominating. This site borders a road to the east, with a suburban sector within a secondary forest dominated by *Ligustrum lucidum* W. T. Aiton (“Evergreen tree”) to the west, south, and north. Site #2 was located at 26°43′ S latitude and 65°22′ W longitude, and 660 m altitude, within the Sierra de San Javier park, a protected wildlife area belonging to Universidad Nacional de Tucumán (UNT). The site is surrounded by buildings belonging to UNT, mixed with disturbed wild vegetation patches. Both sampling sites were 4.7 km apart and located at the foothills of the San Javier Mountain (Figure 1), where the climate is subtropical, with a dry season from May to October and a humid–warm season from November to March, with 21.5 °C and 900 mm of average annual temperature and rainfall, respectively [37].

### 2.2. Drosophilid Puparia Sampling

Drosophilid puparia were collected from three different pupation microhabitats (Figure 2): (a) on the fallen fruit, but inside it, i.e., puparia located in the mesocarp (=flesh), (b) on the fallen fruit, but outside it, i.e., puparia attached to the fruit rind, into shallow external fruit fissures, and largely or partially protruding from the fruit skin, and (c) in the soil, i.e., puparia buried either underneath the fruit or close to it. These three pupation microhabitats will hence forth be named “inside fruit flesh”, “outside fruit flesh”, and “soil”, respectively.

The puparia-collecting procedure involved randomly selecting 20 fallen fruits per sampling date during feral peach and guava fruiting seasons. In this regard, six surveys of drosophilid puparia of peach and six of guava were carried out between December 2016 and January 2017 (early summer) and March 2017 and April 2017 (late summer and early autumn), respectively. Each selected fruit (peach or guava) was removed and examined using a hand-held magnifier with an X20 glass lens at the study site. All drosophilid puparia found inside pulp fruit or attached to the fruit rind were extracted with either a blunt-tip tweezer or a soft-bristled paintbrush. Then, puparia were placed separately into 8 × 5 cm (diameter × height) plastic cups according to the pupation habitat where they were found. Each cup had a thin layer of sterilized, moistened vermiculite Intersum^®^ (Aislater S.R.L., Cordoba, Argentina) on the bottom to avoid desiccation during transport to the lab. Cups were covered with plastic lids with pinholes. In addition, the soil underneath each fruit and the soil sector around the fruit in a 3 cm radius were dug with a hand shovel up to ~2 cm deep to find buried puparia. The extracted soil of each sampled fruit was placed individually in a plastic bag, and its top was closed with a rubber band. Both cups and bags were placed in plastic crates (32 × 24 × 12 cm) and taken to the Pest Biological Control Department (DCBP, Spanish acronym) from the Planta Piloto de Procesos Industriales Microbiológicos y Biotecnología (PROIMI,) in San Miguel de Tucumán, the capital city of the Tucumán province. PROIMI is ~6 and ~10 km away from study sites #1 and #2, respectively.

### 2.3. Drosophilid Puparia Processing and Identification

Each soil sample was sieved through a 1 mm metal-mesh sieve at the DCBP-PROIMI‘s laboratory. Puparia retained in the sieve were removed and then identified, as were puparia from the fallen fruit. *Drosophila suzukii* puparia were differentiated from those of other drosophilids by the external shape of the characteristic anterior spiracles, composed of two tubes with plumose-shaped tips on the top [23,39]. Identified *D. suzukii* puparia were separated from the remaining drosophilid puparia and placed into 5 × 6-cm (diameter × height) disposable clear plastic cups. These cups had sterilized 2 mm-thick vermiculite in the bottom and a plastic lid with pinholes to facilitate internal oxygenation. The vermiculite inside the cups was sprinkled every three days with purified water. Puparia were differentiated according to the habitat from which they were recovered and placed in individualized cups. The same procedure was carried out with other *Drosophila* Fallén puparia, identified as belonging to the *Drosophila melanogaster* species group [40], but not differentiated at the species level. These saprophytic drosophilids will henceforth be referred to as *Drosophila* spp. in the text. All cups were conditioned in a room at 26 ± 1 °C, 80 ± 5% RH, and 10:14 h L:D until adult flies and parasitoids emerged.

### 2.4. Adult Parasitoid and Fly Identification

Drosophilid flies were identified by M.J.B.B., and parasitoid specimens by S.M.O. and Fabiana Gallardo (Facultad de Ciencias Naturales y Museo, Universidad Nacional de La Plata, La Plata, Argentina). Gibson’s [41] and Risbec’s [42] keys were used to identify the pteromalid and the diaprid, respectively. Voucher fly and parasitoid adult specimens were stored at the entomological collection of the Fundación Miguel Lillo, in San Miguel de Tucumán. Parasitoid specimens were also deposited into the entomological collection of the Museo de Ciencias Naturales de la Plata, Buenos Aires, Argentina.

### 2.5. Data Analysis

The response variables analyzed were the drosophilid and parasitoid relative abundance per microhabitat, as well as the parasitism. The drosophilid relative abundance was calculated as the total number of *D. suzukii* or *Drosophila* spp. puparia recovered per microhabitat over the total number of *Drosophila* puparia. Parasitoid relative abundance was calculated as the total number of *P. vindemiae,* or *T. anastrephae* adults that emerged from *D. suzukii* or *Drosophila* spp. puparia per microhabitat over the total number of parasitoid individuals recovered from each drosophilid species. The parasitism percentage was estimated as the number of emerged parasitoids over the number of *D. suzukii* or *Drosophila* spp. puparia recovered from each microhabitat per 100.

The statistical analysis was performed using the software R [43]. For the analysis of drosophilids’ habitat usage, and parasitoid attack, the factorial model for nonparametric analysis of variance, Aligned Rang Transformation ANOVA in the packages ‘*ARTool*’ [44], was performed. First, the algorithm aligned the fixed effects and classified them using the model function, then generated linear models from the transformed data and analyzed the variance using the “*anova.art*” function. A post hoc pairwise comparison (Fisher’s least significant difference = LSD) was conducted to show differences between factor levels using a Bonferroni–Holm adjustment method using the ‘art.con’ function [45]. Kruskal–Wallis tests from the “*Agricolae*” package [46] were performed to determine microhabitat preferences for pupation between saprophytic drosophilids and *D. suzukii*. The library ‘*rcompanion*’ function was used to obtain letters that display the significant difference in figures. Violin box plots are used to show the resulting data from the study. Aside from displaying the summary statistics, using violin box plots of the package “ggplot2” [47] to plot numerical data, the entire data distribution (raw data) is shown (Appendix A).

## 3. Results

### 3.1. Drosophilid Fly Abundance and Relationship with Microhabitats Tested

The abundance of saprophytic drosophilid puparia belonging to the *Drosophila melanogaster* group was two- and four-fold higher than that of *D. suzukii* puparia found on peach and guava, respectively (Figure 3A,B).

Both saprophytic drosophilids and *D. suzukii* were found in all three microhabitats tested, but with remarkable abundance differences. In both host fruit species, *Drosophila* spp. puparia were significantly predominant in the “inside fruit flesh” habitat, while the lowest number of saprophytic drosophilid puparia was found in the soil (Peach, *Χ*^2^ = 295.66, *df* = 2, *p* < 0.0001; Guava, *Χ*^2^ = 295.61, *df* = 2, *p* < 0.0001) (Figure 4A,B). In peach, SWD puparia were slightly more abundant in both “inside fruit flesh” and “outside fruit flesh” microhabitats than in “soil” (*Χ*^2^ = 40.51, *df* = 2, *p* < 0.0001), while in guava, there were no significant differences between the three microhabitats (*Χ*^2^ = 5.11, *df* = 2, *p* = 0.0770) (Figure 4A,B). A comparative analysis of the abundance of saprophytic *Drosophila* spp. and SWD puparia by microhabitat and fruit species showed significant differences between all tested conditions (Table 1). Numbers of *Drosophila* spp. puparia were 4-, 2-, 6-, and 3.5-fold considerably higher than those recorded for SWD from both “inside fruit flesh” and “outside fruit flesh” microhabitats, in both peach and guava, respectively (Figure 4A,B).

Similarly, the number of SWD puparia found in the soil beneath or near fruit increased by four- and two-fold, significantly higher than that recorded for *Drosophila* spp. in peach and guava tree-dominated environments, respectively (Figure 4A,B).

### 3.2. Pupal Parasitoid Abundance and Relationship with Microhabitats Tested

The only pupal parasitoid species associated with drosophilids in both host fruits were *T. anastrephae* and *P. vindemiae*. The former species was 1.8-fold more abundant than the second one. The number of *T. anastrephae* specimens recovered from saprophytic *Drosophila* spp. was three- and seven-fold higher than that recorded from SWD in peach and guava, respectively (Figure 5A,B). The number of *P. vindemiae* specimens recorded from SWD puparia was slightly higher, 1.3- and 1.4-fold, than that obtained from saprophytic *Drosophila* spp. puparia in peach and guava, respectively (Figure 5A,B). When the success of both *T. anastrephae* and *P. vindemiae* in the parasitizing puparia of both *Drosophila* spp. and *D. suzukii* in peach and guava was tested, significant differences were recorded for both categorical factors, such as the type of microhabitat used for host parasitization and the parasitized drosophilid species, as well as their interaction (Table 2).

*Trichopria anastrephae* was remarkably successful in parasitizing saprophytic *Drosophila* spp. puparia located inside peach and guava flesh, followed, in decreasing order, by puparia found “outside fruit flesh” and in the “soil” (Figure 6A,B). The above pattern was also recorded in SWD puparia (Figure 6A,B). However, *T. anastrephae* was considerably more successful in parasitizing *Drosophila* spp. puparia than SWD puparia in both “inside fruit flesh” and “outside fruit flesh” microhabitats; this was not the case for puparia located in the soil, as there was no significant difference in parasitism between drosophilids (Figure 6A,B). *Pachycrepoideus vindemiae* significantly parasitized more saprophytic *Drosophila* spp. puparia outside fruit flesh when compared to the other tested microhabitats on peach and guava (Figure 6C,D). Nevertheless, the parasitism success of *P. vindemiae* on SWD puparia located in both “outside fruit flesh” and “soil” was statistically similar to that recorded from *Drosophila* spp. puparia outside fruit flesh when only peach was evaluated (Figure 6C). In guava, significantly more SWD puparia were also parasitized by *P. vindemiae* in both “outside fruit flesh” and “soil” microhabitats, but the success of such parasitism was statistically lower than that recorded from *Drosophila* spp. puparia located outside the flesh (Figure 6D). *Pachycrepoideus vindemiae* parasitized a significantly higher number of *Drosophila* spp. puparia located “inside fruit flesh” than SWD puparia found in the same microhabitat in peach, although there were no statistical differences for guava (Figure 6C,D).

## 4. Discussion

The success of biological control programs involving parasitoids relies, among many factors, on the knowledge of the resident parasitoid assemblage associated with the invading pest and, crucially, on understanding the mechanisms that allow the coexistence of competing parasitoid species. This sort of ecological insight provides a better understanding of the impact exerted by each resident parasitoid species on the target pest population. Furthermore, all such information is critical for developing and implementing biological control strategies, including exotic or local parasitoid species. In this framework, results of the field study carried out in the fruit-growing province of Tucumán, northwestern Argentina, evidenced niche partitioning as a mechanism involved in facilitating the coexistence of two resident generalist parasitoids, *T. anastrephae* and *P. vindemiae*, attacking puparia of both the invasive fruit pest *D. suzukii* and local saprophytic drosophilid species in the same non-crop fallen fruit. In particular, results revealed interesting aspects of the fruit–drosophilid–parasitoid trophic relationship: (1) differentiated patterns of drosophilid puparia distribution in microhabitats; (2) proportions of *P. vindemiae* adults recovered from SWD puparia that are higher than or similar to those found for *Drosophila* spp. puparia from the *D. melanogaster* group; (3) a discernible trend of *P. vindemiae* females to focus their attack on the puparia of both SWD and resident saprophytic *Drosophila* spp. outside the fleshy inner of the dropped fruit; (4) a strong preference of *T. anastrephae* females for targeting puparia of resident saprophytic drosophilid species, particularly those located inside the fruit.

Two issues should be emphasized concerning the first finding. Firstly, a markedly higher abundance of resident drosophilid puparia over SWD puparia in both host fruits surveyed was observed, but this difference was more evident in guava. Secondly, there was increased resident drosophilid puparia in both inside and outside peach and guava flesh, whereas SWD puparia prevailed in the soil relative to all the other drosophilids. The above is consistent with the prediction based on space-use patterns between the invasive drosophilid species and the local ones. These differences may be related to the fruit ripeness stage preferred by SWD females for egg laying. Rather than overripe, fallen and damaged fruit, these females choose ripe, fresh fruit still on the plant [15]. Therefore, the SWD female exploits mostly fruit in the ripening stages, due to their availability for other *Drosophila* species [31]. In turn, the mature larvae usually tend to migrate from the fruit hanging on the branch, in order to pupate in the soil [48,49]. Thus, the preference for healthy fruits enables the SWD female to exploit novel niches by avoiding competition with other drosophilids [14]. However, SWD females may sometimes lay their eggs in fallen, wounded, and/or fermenting fruit, in situations involving a shortage or non-availability of suitable hosts [50,51]. The females of the *Drosophila melanogaster* group (e.g., *D. melanogaster* and *D. simulans*) are saprophytic flies; they feed and oviposit on damaged, decaying, or fermenting fruits [52], and their larvae usually pupate in the dropped fruit, covering a significant part of their biological cycle in the same microenvironment, in contrast to the standard SWD female oviposition behavior. Given these differences in fruit ripeness preference, the female of resident drosophilids usually oviposits on the host at a later stage than the SWD female. Thus, SWD larvae may complete their development first, and mature larvae usually drop out of the fruit. Therefore, another critical factor to consider is the interspecific larval competition for resources between SWD and saprophytic drosophilids on damaged or already rotten fruit. Such competition may cause a decrease in SWD population growth [50], which might also influence the lower density of SWD puparia inside the host fallen fruit versus resident drosophilid puparia. However, a higher proportion of SWD puparia were found in the fruit than buried around the fallen fruit. Although the SWD mature larva tends to leave the fruit, it can pupate inside the ripe fruit, as [31] demonstrated using infested cherries under lab conditions. These authors found more SWD larvae pupated in the cherry fruit than in the soil. However, recent laboratory trials, carried out by the senior author of the current study, revealed that 60–70% of the total SWD larvae pupated in an artificial pupation medium close to infested peaches, while 75–80% of total *D. melanogaster* puparia remained on fruits. In those lab trials, healthy, ripe, soft peaches were offered to 20 mated SWD females for 24 h. The fruit was then kept in the experimental cage, and on the fifth day, when the fruit showed spotted and rotting sectors, the 20 mated *D. melanogaster* females were released and remained in the cage for 24 h. High numbers of drosophilid larvae left the fruit 1–3 days after *D. melanogaster* females oviposited in peaches. It was then verified by puparium identification that all those larvae were *D. suzukii*. Overall, under competitive interspecific conditions, SWD females are specialized to oviposit on healthy fruit, which is highly preferred, although these females can be flexible to use wounded and fermented fruit as well [50].

The second finding showed an apparently closer trophic relationship between *P. vindemiae* and SWD than with saprophytic drosophilid species in the two natural environments studied. This trait was even more striking when puparia recovered from fallen peaches were analyzed. In this regard, SWD puparia were parasitized by *P. vindemiae* 1.4 times more than by *T. anastrephae*, as opposed to saprophytic drosophilid puparia, which were 3.3-fold more parasitized by *T. anastrephae*. In guava, the incidence of *P. vindemiae* affecting Drosophilidae populations was lower than that recorded in peach. The above was evidenced by the lower proportion of *P. vindemiae* adults recovered from puparia taken from guava, compared to *T. anastrephae* adults. The *P. vindemiae* and *T. anastrephae* adult ratios recorded from SWD puparia were close to 1:1, while for resident drosophilid puparia, there were nearly six *T. anastrephae* individuals per *P. vindemiae* adult. This remarkable difference in the proportion of *P. vindemiae* adults recovered from Drosophilidae puparia between both sampling environments may be due to two related events. Firstly, there is a higher predominance of saprophytic drosophilids on SWD on guava than on peach. In this framework, the proportion of resident drosophilid puparia over SWD was twice higher in guava than in peach. Secondly, there is a low natural population of *P. vindemiae* relative to that of *T. anastrephae* in the guava tree-dominated habitat. In this regard, data from direct field observations at the surveying sites recorded an average of 4.2-fold more *P. vindemiae* adults in the peach tree-dominated wild forest compared to the guava-sampling site. Field observations were made every 15 m for 2 h on each sampling date in both natural environments. Regardless of the above two events, *P. vindemiae,* relative to *T. anastrephae,* is a resident parasitoid, mostly predominant on SWD puparia in both tested habitats. These results support data recently published by Buonocore Biancheri et al. [16], which point to *P. vindemiae* as an attractive agent of SWD natural mortality in disturbed wild environments from the province of Tucumán. However, field studies in southern Brazil showed a higher prevalence of *T. anastrephae* in SWD puparia than that of *P. vindemiae* [35,53]. Analogous data were published on *T. drosophilae* naturally attacking SWD puparia in Europe [54]. However, those contrasts between the results of field studies conducted mainly in Tucumán and southern Brazil may be due to a wide range of biotic and abiotic factors. These may include the disturbance characteristics of the study environment, the variation in the population density of the target pest and competitive frugivorous species, weather conditions during the sampling period, and the host fruit species and its abundance.

All the above factors notably influence the composition of the parasitoid assemblage associated with the target pest and the abundance of each resident parasitoid species [55,56]. However, laboratory establishment of the population lines of both pupal parasitoids tested in the current field study is foreseen as a later research step. In this context, studies with Brazilian population lines of *T. anastrephae* and *P. vindemiae* lab-reared on SWD puparia showed a significant prevalence of the former parasitoid species over the latter in terms of parasitism and adult emergence rates [25]. Similar findings were reported by Wang et al. [31], who found a strong predominance of lab-reared *T. drosophilae* over lab-reared *P. vindemiae* as a parasitoid of both *D. suzukii* and *D. melanogaster*. Similarly, Daane et al. [57] and Wang et al. [58] pointed to *T. drosophilae* as a more efficient SWD parasitoid in laboratory tests than other well-known pupal parasitoids, such as *P. vindemiae*. Likewise, Wolf et al. [59] found in the combined release of *T.*
*drosophilae* and *P.*
*vindemiae* under semi-field experiments that almost all the parasitoid offspring that emerged from SWD puparia were *T. drosophilae* adults, despite the two microhabitats tested: soil and foliage.

Under this framework, several authors pointed to *T. drosophilae* as the pupal parasitoid species with the highest potential for SWD control [60,61,62,63,64,65]. Evidence was provided through augmentative releases of *T. drosophilae* on cherries, in either crop or non-crop areas in southern Trento, Italy, which achieved a 34% reduction in fruit infestation by *D. suzukii* in unmanaged vegetation areas surrounding orchards [20]. Similarly, mass releases of *T. drosophilae* in berry crops at Colima and Jalisco, Mexico, reduced SWD populations by 50 to 55% [66]. Given the contrast between the field findings of the current study and those from both laboratory [25,31] and semi-field [59,61] studies, it is relevant to assess Argentinian population lines of both pupal parasitoids under lab conditions. This would enable a comparative assessment of *P. vindemiae* and *T. anastrephae* as biocontrol agents of *D. suzukii* by determining host preference, regulating offered host densities, and analyzing the interspecific competition.

The third finding showed that *P. vindemiae* was the predominant parasitoid species recovered from Drosophilidae puparia that was externally attached to the fruit skin, enclosed in outer fruit injuries, protruding from the fruit rind, or directly buried under the fallen fruit. On the contrary, results of the fourth finding revealed that the highest levels of *T. anastrephae* adult abundance were mainly recorded from resident drosophilid puparia sampled directly from peach and guava flesh. Such data were reflected when adult proportions of both pupal parasitoid species recorded from the different tested habitats were comparatively assessed. On this issue, around seven and five *P. vindemiae* adults per *T. anastrephae* adult were recovered from SWD puparia collected from peach and guava, respectively, in the two habitats not involving fruit flesh. In turn, about twice as many *P. vindemiae* adults per *T. anastrephae* adult were recorded from resident drosophilid puparia collected from “outside fruit” and “soil” microhabitats in both host fruit species. In connection with the second finding, the above data support *P. vindemiae*’s prevalence over *T. anastrephae* on SWD puparia. Although saprophytic drosophilid puparia found in both “outside fruit” and “soil” microhabitats yielded relatively more *P. vindemiae* than *T. anastrephae* adults, the highest *P. vindemiae* adult abundance was recorded from SWD puparia. This result was even more evident when only SWD puparia sampled from the “soil” were considered. About 92% of the total pupal parasitoid adults recovered from SWD puparia sampled from the “soil” around peach fruit were *P. vindemiae*. In the same way, all SWD puparia collected from soil in the guava-dominated environment yielded only *P. vindemiae* adults. Th soil was the most favorable microhabitat for *P. vindemiae* females to parasitize Drosophilidae puparia. The *P. vindemiae* female may tend to forage in this type of microhabitat rather than inside the fruit. This assertion may also be corroborated by data on the *P. vindemiae*: *T. anastrephae* adult ratio recorded from resident drosophilid puparia found in soil. At least 35- and 11-fold more *P. vindemiae* than *T. anastrephae* adults were recovered from those buried puparia in guava and peach sampling sites, respectively. *Pachycrepoideus vindemiae* either rarely frequented the inside of the fruit or found it difficult to parasitize host puparia in this microhabitat, showing the lowest levels of abundance recorded in both tested fruit species. Thus, *P. vindemiae* adults recovered from either SWD or resident drosophilid puparia found inside guava and peach flesh only accounted for 6–20% of all individuals of this species. Interestingly, previous studies in Switzerland under both semi-field [59] and open-field [36] conditions demonstrated a preference of *P. vindemiae* for parasitizing drosophilid puparia in the foliage, while *T. drosophilae* mostly parasitized host puparia on the ground. The current study focused on sampling drosophila puparia on the soil and those associated with fallen fruit, without considering damaged fruit located in the plant canopy. However, the forthcoming surveys of drosophila parasitoids will cover wounded or rotting fruit still located in the canopy.

Based on the highly contrasting *P. vindemiae* parasitism data between microhabitats tested in the current study, the interference with *T. anastrephae* was more likely a critical factor influencing *P. vindemiae* performance. That is, given a competitive interaction with *T. anastrephae* for the resource, *P. vindemiae* probably faces a disadvantageous situation. Laboratory trials previously reported by da Costa Oliveira et al. [25] showed that *T. anastrephae* from the Brazilian population lineage was competitively superior to *P. vindemiae*, and achieved substantially higher levels of parasitism in SWD puparia when the two parasitoids interacted with each other. Similar results were also reported in interspecific competition studies between *T. drosophilae* and *P. vindemiae* under lab conditions [31] or in semi-field trials [59], where *P. vindemiae* only achieved the highest parasitism when released alone. Both *T. anastrephae* and *P. vindemiae* can discriminate hosts previously parasitized by the other species [25], being relevant for their females to oviposit first on the typical host. Usually, the first parasitoid species ovipositing into the host prevails in an intrinsic competition [67]. However, *T. anastrephae* as *T. drosophilae* [31] may have a set of biological features that allow it to out-compete *P. vindemiae*. These *T. anastrephae* traits may include the following: (a) faster embryonic development, (b) first-instar larvae better being equipped (larger mandibles and fast movements) for encountering competitors, and (c) higher foraging efficiency, which involves less time spent handling the host due to a higher mature egg load.

It is also worth noting that *T. anastrephae* was the dominant parasitoid species recovered from host puparia found inside the fruit in the current study. Hence, it is likely that *T. anastrephae* females preferentially foraged in this microhabitat. About 73% of all *T. anastrephae* adults recovered from resident drosophilid puparia in feral peach and guava were from those sampled directly inside the mesocarp. Interestingly, results also revealed that most of the *T. anastrephae* adults associated with *D. suzukii* (over 56%) were from puparia collected from inside guava or peach fruit. This information is consistent with da Costa Oliveira et al. [25], who stated that *T. anastrephae* females of the Brazilian population lineage successfully parasitize SWD puparia inside strawberry fruits. In addition to Drosophilidae puparium survey data, field records through direct inspection inside the fruit evidenced an average proportion of 16.5 *T. anastrephae* adults per each *P. vindemiae* adult, in this microhabitat by testing 36 fruits (18 guavas and 18 peaches) during all six collecting dates (three fruit of each species per sampling date). A comparative laboratory study between *P. vindemiae* and *T. drosophilae* showed that the diaprid was more effective than the pteromalid for attacking SWD, and parasitism by either parasitoid species was higher in puparia located on cherry fruit, rather than in the soil [31]. This study also showed a slight preference for *T. drosophilae*, similar to *T. anastrephae*, for attacking host puparia on fruits, although Wang et al.’s [31] work was, methodology-wise, different from the current study.

Such information would provide evidence of a resource and niche partitioning, probably aimed at reducing or avoiding interspecific competition between resident pupal parasitoid species. Initially, the above-discussed data plus the second finding outcome would reflect a differentiated use of available resources in the surveyed environments, i.e., different drosophilid species as hosts. This background would mainly display *P. vindemiae* females parasitizing *D. suzukii* puparia and *T. anastrephae* females mostly attacking saprophytic drosophilid puparia. These host preference assertions for *P. vindemiae* and *T. anastrephae* might be supported by differences in the co-evolutionary history between the parasitoid and its host. On this basis, *T. anastrephae* is a neotropical-native parasitoid species [29,33] that has co-evolved in sympatry with saprophytic drosophilid species, such as those of *D. melanogaster* group [16]. Thus, a close trophic association occurs between *T. anastrephae* and non-pest saprophytic drosophilids, whereas with *D. suzukii,* a new trophic association has recently been established, which is naturally uncommon, due to the incidence of preferred hosts. In contrast, *P. vindemiae* is a worldwide cyclorrhaphous dipteran parasitoid that was introduced in several Latin American countries as a biocontrol agent against tephritid pests [27]. Although its first record in Argentina dates back to the 1940s, it is most likely an exotic parasitoid species [68]. Therefore, the high level of polyphagy associated with the lack of a common co-evolutionary history with saprophytic drosophilid species in northwestern Argentina supports a closer trophic association between *P. vindemiae* and *D. suzukii*. Subsequently, data from the current study suggest an apparent preference for *P. vindemiae* for parasitizing Drosophilidae puparia in microhabitats mostly exposed to female parasitoid attacks. In line with this statement, the *P. vindemiae* female would exhibit a foraging behavior targeted mainly to host puparia in the soil. Host puparia buried beneath or adjacent to fruit likely provide a *T. anastrephae*-free microhabitat, which would ease *P. vindemiae* females’ foraging for the host in the soil, without interference from the closest competitor. This scenario points to probable niche segregation between both pupal parasitoid species at a spatial scale. This means *T. anastrephae* females focuse on foraging mostly inside the fruit for host puparia, whereas *P. vindemiae* females target their host search effort in habitats occasionally frequented by the competitor, such as both “outside fruit flesh” and “soil”. As pointed out by Wang et al. [31], when discussing the *T. drosophilae*–*P. vindemiae* competitive relationship, an alternative host does appear to reduce interspecific competition between such species, although these parasitoids showed no preference for *D. suzukii* or *D. melanogaster* when tested in the laboratory. However, in natural conditions, sympatric species tend to reduce competition by using different resources or habitats [3].

In conclusion, results reveal that both divergence in host choice and spatial patterns of same-resource preference among potential competitors, such as *P. vindemiae* and *T. anastrephae*, may mediate the coexistence of these two pupal parasitoids species in each natural environment tested in the current study. Given the apparent preference of the native *T. anastrephae* for resident saprophytic dipteran puparia, mainly located in guava or peach fruit, *P. vindemiae* might be more suited to forage in less competitive microhabitats, such as in the soil or outside of the fruit flesh, in which puparia of the exotic *D. suzukii* would naturally prevail in these habitats. From a SWD management approach, this scenario suggests that both pupal parasitoids have potential as *D. suzukii* biological control agents. This is because such niche partitioning primarily involves differentiated exploitation of host microhabitats, influencing the efficiency of both parasitoids in suppressing *D. suzukii* populations. Such an approach regarding the use of both pupal parasitoid species, based on a differentiation in host microhabitat preference (soil vs. foliage), was highlighted by Wolf et al. [59] relying on semi-field study results in Switzerland. Likewise, Kruitwagen et al. [69] and Jarret et al. [70], in studies based on experimental adaptation studies of resident parasitoids to the invasive *D. suzukii*, pointed out that both *T. anastrephae* and *P. vindemiae* might offer a greater potential to control SWD natural populations over larval parasitoids. Consequently, combining the two resident parasitoid species in wild non-crop environments may be an advisable alternative for local SWD management, either through augmentative releases [20] or through a conservation biological control program [71]. It is worth analyzing this initiative from an area-wide SWD management approach, as suggested by Garcia et al. [11], Rossi-Stacconi et al. [20], Garcia [21], Wang et al. [22]. In this context, parasitoid releases should mainly be performed in wild areas, where known, non-crop, alternative SWD hosts are abundant and may increase the risk of SWD infestations in surrounding fruit crops [20,22,72]. Furthermore, pupal parasitoids would be more effective if released early in the fruiting season, when SWD numbers are still low, to avoid the pest population increase [20,59,72]. This SWD biological control strategy is particularly relevant for the province of Tucumán, where feral guavas and peaches share the same geographical space with commercial berry orchards, as Tucumán hosts most of the soft fruit crops in fruit-growing regions of northern Argentina [73]. Both feral fruit species allow the sustainability of SWD populations during the season in which commercial berry crops are not in production, representing a high economical risk for the local fruit industry. In this context, the use of both the studied parasitoids is a practical and useful alternative for berry growers in Tucumán; they may release them in areas of wild vegetation adjacent to their crops or in orchards or backyards where there is no phytosanitary control. Finally, it is relevant to examine whether that niche differentiation in both parasitoid species occurs in other fruit host species, such as berries, or in other natural environments, such as berry crops in the outlying areas surrounding crops.

## Figures and Tables

**Figure 1 insects-14-00222-f001:**
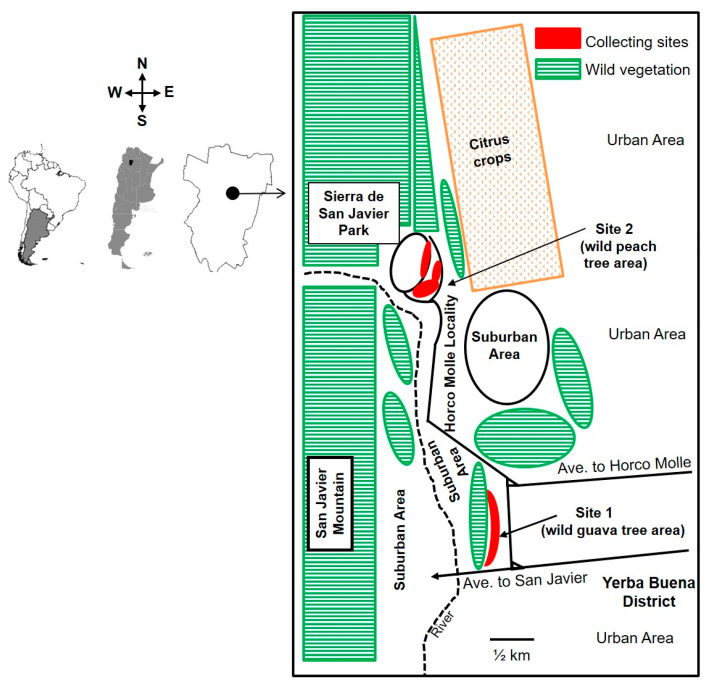
Study area showing the two sampling sites: Site 1, a guava tree-dominated habitat; Site 2, a peach tree-dominated habitat. The area is located in Horco Molle, Yerba Buena district, Tucumán province, northwestern Argentina.

**Figure 2 insects-14-00222-f002:**
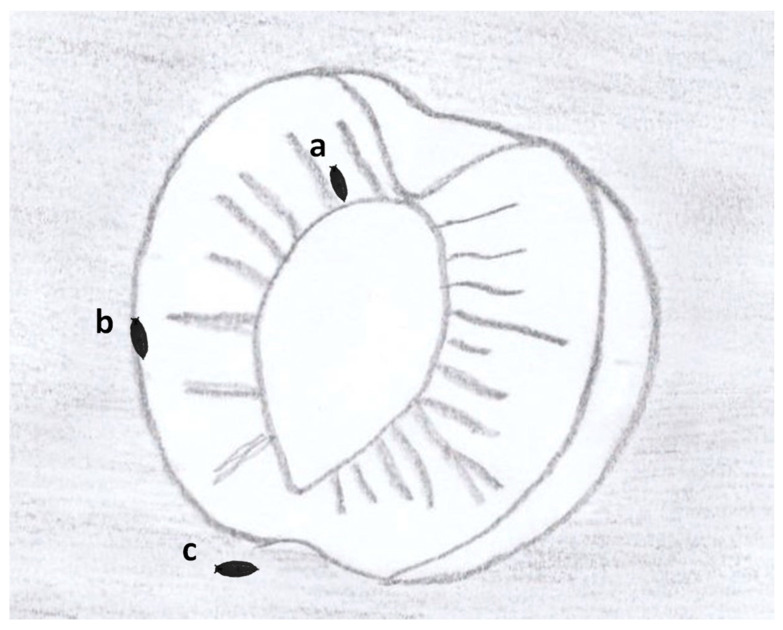
Scheme of a peach fruit fallen on the soil showing three microhabitats, in which the larvae of both saprophytic *Drosophila* spp. and *Drosophila suzukii* can pupate: (**a**) “inside fruit flesh”, (**b**) “outside fruit flesh” and (**c**) “soil”.

**Figure 3 insects-14-00222-f003:**
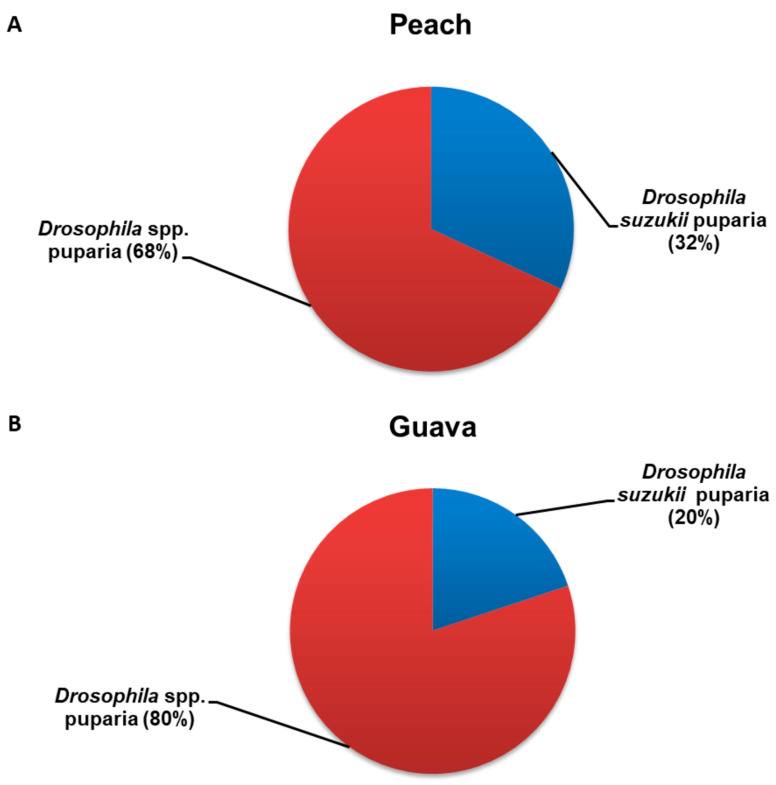
Relative abundance of both *Drosophila suzukii* and saprophytic *Drosophila* spp. puparia recovered from (**A**) peach and (**B**) guava.

**Figure 4 insects-14-00222-f004:**
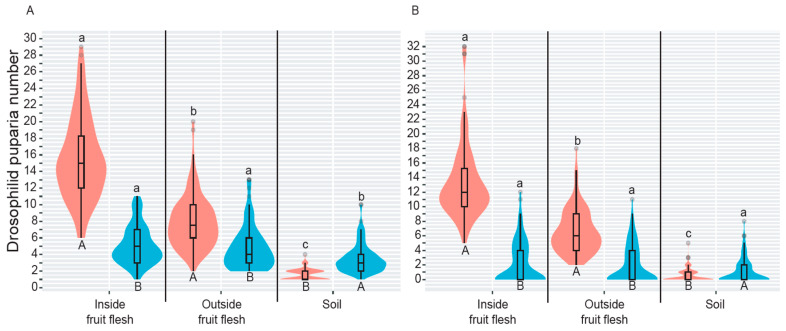
Violin-box plots showing the kernel probability density of a drosophilid (saprophytic *Drosophila* spp. from *D. melanogaster* group = red, and *Drosophila suzukii* = blue) choosing a specific microhabitat in peach and in guava. Violin-box plots showing the kernel probability density. Different letters show significant differences at *p* = 0.05 (LSD test with the Bonferroni–Holm adjustment method). To compare all microhabitats, lowercase, and uppercase letters are used, respectively, for *Drosophila* spp. and *Drosophila suzukii*. The rectangular white bar in the center of the violin box and the black horizontal line inside the bar show the interquartile range and the median, respectively; the black vertical lines stretched from the bar show the lower/upper adjacent values, while black dots display the outlier data.

**Figure 5 insects-14-00222-f005:**
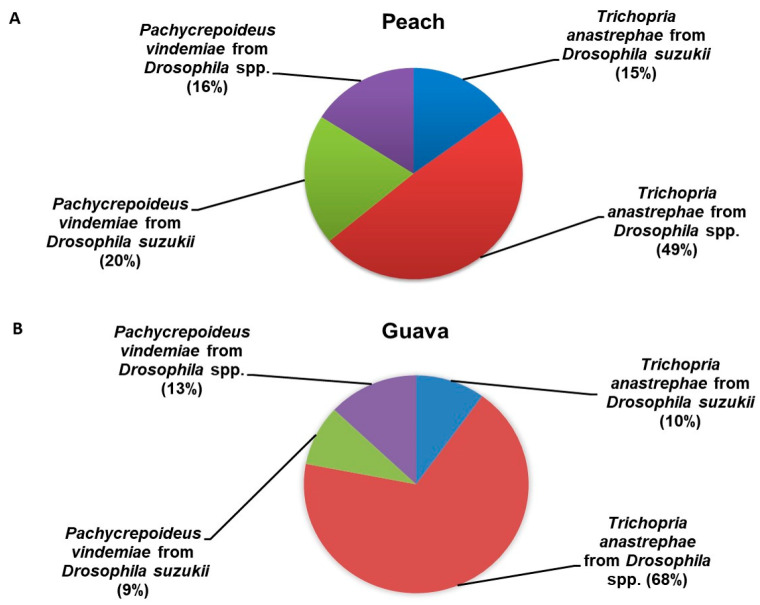
Relative abundance of *Pachycrepoideus vindemiae* and *Trichopria anastrephae* adults recovered from *Drosophila suzukii* and saprophytic *Drosophila* spp. Puparia, associated with (**A**) peach and (**B**) guava.

**Figure 6 insects-14-00222-f006:**
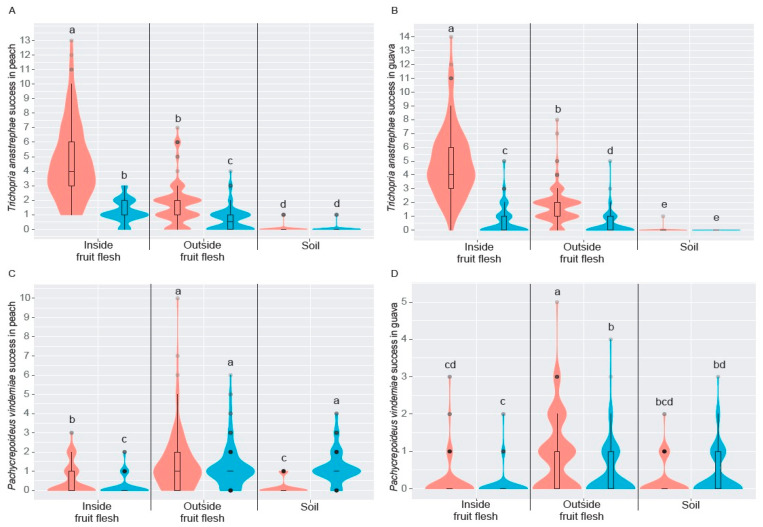
Violin-box plots showing the kernel probability density that drosophilids (saprophytic *Drosophila* spp. from *D. melanogaster* group = red, and *Drosophila suzukii* = blue) are parasitized in a different microhabitat by (**A**) *Trichopria anastrephae* in peach, (**B**) *Trichopria anastrephae* in guava, (**C**) *Pachycrepoideus vindemiae* in peach, and (**D**) *Pachycrepoideus vindemiae* in guava. Different letters show significant differences at *p* = 0.05 (LSD test with the Bonferroni–Holm adjustment method). The rectangular white bar in the center of the violin box and the black horizontal line inside the bar show the interquartile range and the median, respectively; the black vertical lines stretched from the bar show the lower/upper adjacent values, while black dots display the outlier data.

**Table 1 insects-14-00222-t001:** Summary of the Kruskal–Wallis test on puparia abundance comparison between saprophytic *Drosophila* spp. and *D. suzukii* recorded by microhabitat and host fruit species.

Microhabitat	Host Fruit	Statistical Analysis Outcome		
		*df*	*Χ* ^2^	*p*
Inside fruit flesh	Peach	1	166.94	<0.0001 *
	Guava	1	173.95	<0.0001 *
Outside fruit flesh	Peach	1	66.78	<0.0001 *
	Guava	1	108.91	<0.0001 *
Soil	Peach	1	122.78	<0.0001 *
	Guava	1	0.02	=0.0270 *

* Significant variables.

**Table 2 insects-14-00222-t002:** Summary of Aligned Rank Transform ANOVA on the effect of the type of microhabitat used for host parasitism (=THU), the parasitized drosophilid species (=PDS), and their interaction on the adult emergence of both *Trichopria anastrephae* and *Pachycrepoideus vindemiae*, with data recorded from saprophytic *Drosophila* spp. and *D. suzukii* puparia recovered from peach and guava.

Host Fruit	Parasitoid Species	Source of Variation	Statistical Analysis Outcome			
			*df*	Residuals *df*	*F*	*p*
Peach	*T. anastrephae*	THU	2	714	566.19	<0.0001 *
		PDS	1	714	389.58	<0.0001 *
		THU × PDS	2	714	204.23	<0.0001 *
Guava	*T. anastrephae*	THU	2	714	599.24	<0.0001 *
		PDS	1	714	733.73	<0.0001 *
		THU × PDS	2	714	298.26	<0.0001 *
Peach	*P. vindemiae*	THU	2	714	85.728	<0.0001 *
		PDS	1	714	84.386	<0.0001 *
		THU × PDS	2	714	113.08	<0.0001 *
Guava	*P. vindemiae*	THU	2	714	52.673	<0.0001 *
		PDS	1	714	102.91	<0.0001 *
		THU × PDS	2	714	42.630	<0.0001 *

* Significant variables.

## Data Availability

The data presented in this study are available in Appendix A here.

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
