# Peer review of "Implications of the Niche Partitioning and Coexistence of Two Resident Parasitoids for Drosophila suzukii Management in Non-Crop Areas"

_insects, 2023, doi:10.3390/insects14030222_

Round 1
Reviewer 1 Report
Dear Dr. Ovruski,
your manuscript titled "Implications of the niche partitioning and coexistence of two resident parasitoids for D.s. management in non-crop areas" presents very interesting and new data. Both pupal parasitoids studied are the most promising candidates for biological control of D. suzukii in various habitats and newly invaded regions where this drosophilid is an important fruit pest.
The manuscript is well written, methods are clearly described, results adequately and cleary presentend and statitically analyzed.
The discussion is exhaustive and also well written.
Thus, I have only minor comments that I consider would improve the quality of your manuscript:
Material and methods
2.4. Parasitoid and fly identification
Please add the citation of all or the most important taxonomic keys you used for these identifications.
References cited:
Please extend this section - and correspondingly a bit also your introduction and/or discussion - and include recent relevant European puplications:
for example:
Wolf et al. 2020. Pest Manag Sci 77
Wolf et al. 2020. J Pest Sci 93
Rossi Stacconi et al 2018 Biol Control 117
Tivellone et al. 2020. Insects 11
Moreover, with regard to your title, as a reader I would expect a more in-depth discussion of the potential interest of your findings and observations of parasitoid abundance, niche partitioning on management of D. suzukii in commercial peach and guava orchards.
Are those fruits generally cultivated together? In adjacent orchards? What about fruiting and harvesting time periods of these two species ? How does that affect D. suzukii migration into the orchards? How large is fruit damage and infestation rate in those cultivated fruits? Are there any data on augmentative releases of parasitoids in those fruits in Argentina or other countries? Or concerning any other fruit host of this fly?
What about the results of parasitoid releases against D. suzukii in Italy? Is there anything that could be of interest for Argentinian fruit growers?
Also, please explain why you chose non-crop hosts for your study and what that implies for cultivated hosts of the same plant species.
In summary, since you combine ecology and management in your title, please do so in a more detailed way also in your discussion.
Author Response
Reviewer #1. Comments:
1.-Dear Dr. Ovruski, your manuscript titled "Implications of the niche partitioning and coexistence of two resident parasitoids for D.s. management in non-crop areas" presents very interesting and new data. Both pupal parasitoids studied are the most promising candidates for biological control of D. suzukii in various habitats and newly invaded regions where this drosophilid is an important fruit pest. The manuscript is well written, methods are clearly described, results adequately and cleary presentend and statitically analyzed. The discussion is exhaustive and also well written. Thus, I have only minor comments that I consider would improve the quality of your manuscript.
Response. We appreciate the reviewer's very gratifying comments on our study.
2.-Material and methods. 2.4. Parasitoid and fly identification. Please add the citation of all or the most important taxonomic keys you used for these identifications.
Response. We have incorporated the following text in the corresponding section of Materials and Methods: “Gibson's (2020) and Risbec's (1950) keys were used to identify the pteromalid and the diaprid, respectively”. The references included are:
-Gibson, G.A.P. 2000. Illustrated key to the native and introduced chalcidoid parasitoids of filth flies in America north of Mexico (Hymenoptera: Chalcidoidea). http://res2.agr.ca/ecorc/apss/chalkey/keyintro.htm. October, 2000.
-Risbec, J. (1950). La faune entomologique des cultures au Sénégal et
au Soudan: II. Contribution à l’étude des Proctotrupidae
(Serphiidae). Gouvernement général de l’Afrique occidentale française,
[s.l.], (Travaux du Laboratoire d’Entomologie du Secteur Soudanais de Recherche Agronomique – Station de Bambey), 638 p.
3.-References cited. Please extend this section - and correspondingly a bit also your introduction and/or discussion and include recent relevant European puplications, for example: Wolf et al. 2020. Pest Manag Sci 77, Wolf et al. 2020. J Pest Sci 93, Rossi Stacconi et al 2018 Biol Control 117, Tivellone et al. 2020. Insects 11.
Response. The bibliographic references provided by the reviewer #1 are very appropriate for our work; they were included in different new paragraphs in both Introduction and Discussion sections, as suggested by the reviewer. Here, the name of the first author and year of publication are mentioned, but in the revised version of the manuscript, which was submitted, the corresponding number of the reference list appears in the text. The new sentences/paragraphs including bibliographic references are stated mainly in item 4, as they answer the reviewer's queries. However, some sentences/paragraphs in the text include the references listed by the reviewer, below:
In Introduction section:
- “In this regard, information on wild host fruit status on which SWD populations may increase is critical to support management strategies, particularly in wilderness environments surrounding commercial fruit crops (Lee et al. 2015, Kenis et al. 2016, Rossi-Stacconi et al. 2019)”.
-“Microhabitat differentiation was addressed for host puparia sampling based on the strong influence of the microhabitat type, e.g. soil or canopy, on the parasitoid assemblage associated with saprophytic drosophilids consuming decaying organic matter rather than habitat type (Trivellone et al. 2020)”.
In Discussion section:
-“Likewise, Wolf et al. (2021) found in the combined release of T. drosophilae and P. vindemiae under semifield experiments almost all the parasitoid offspring emerged from SWD puparia were T. drosophilae adults, despite the two microhabitats tested, soil and foliage”.
-“Similar results were also reported in interspecific competition studies between T. drosophilae and P. vindemiae under lab conditions [28] or in semifield trials [Wolf et al. 2021], where P. vindemiae only achieved the highest parasitism when released alone”.
4.-Moreover, with regard to your title, as a reader I would expect a more in-depth discussion of the potential interest of your findings and observations of parasitoid abundance, niche partitioning on management of D. suzukii in commercial peach and guava orchards.
--4.a.-Are those fruits generally cultivated together?, In adjacent orchards?, What about fruiting and harvesting time periods of these two species ?, How does that affect D. suzukii migration into the orchards?, How large is fruit damage and infestation rate in those cultivated fruits?.
--4.b.-Are there any data on augmentative releases of parasitoids in those fruits in Argentina or other countries? Or concerning any other fruit host of this fly?. What about the results of parasitoid releases against D. suzukii in Italy?. Is there anything that could be of interest for Argentinian fruit growers?.
Response. 4.a. The reviewer's queries are answered in a new paragraph included in the Introduction section, as follows: “The SWD is found in 64 host plants in 25 families in Latin America. Although most hosts are exotic in this region, about 39% are native plants that can become alternative hosts and reservoirs of the pest in the intercrop period [11]. In Argentina, 15 fruit species have been recorded as hosts of SWD, including both feral guava (Psidium guajava L.) (Myrtaceae) and feral peach [Prunus persica (L.) Stokes] (Rosaceae) [Garcia et al. 2022, Buonocore Biancheri et al. 2022]. These SWD host plants are among the most common and widespread exotic feral fruit species growing in wild vegetation patches adjacent to commercial fruit crops in the northwestern Argentina. Natural infestation levels by SWD in feral guava and feral peach ranged between 5 and 10% per kg of sampled fruit (Escobar et al. 2018; Buonocore Biancheri et al. 2022). The guava is not commercially grown in Argentina. It can only be found as an ornamental plant in gardens or as a backyard fruit tree or scattered in wilderness areas with high levels of human disturbance. The peach is cultivated in northwestern Argentina at a very low scale with no influence in the local or national supply. Peaches are mainly grown in the central-western and south-northeastern regions of Argentina. However, cultivated peaches were not reported to be infested by the SWD, although there are only some records of SWD adults caught in liquid traps placed inside the commercial peach orchard. In spite of the SWD infesting cultivated peach in some Asian, European and North American countries, but in fruits mainly with previous wounds, it is not a natural host of this pest (Kirschbaum et al. 2020)”.
Response. 4.b. To date, no Augmentative releases of parasitoids against D.suzukii have been made in Argentina. References to those papers on augmentative releases of Trichopria drosophilae in Italy and in Mexico are included in the new version of the manuscript (Discussion section) as follows: “Under this framework, several authors pointed to T. drosophilae as the pupal parasitoid species with the highest potential for SWD control (Mazzetto et al. 2016; Kaçar et al. 2017; Rossi-Stacconi et al. 2018; Pfab et al. 2018; Wang et al. 2018; Yi et al. 2020). Evidence was provided through augmentative releases of T. drosophilae on cherries in either crop or non-crop areas in southern Trento, Italy, which achieved a 34% reduction in fruit infestation by D. suzukii in unmanaged vegetation areas surrounding orchards (Stacconi et al. 2019). Similarly, mass releases of T. drosophilae in berry crops at Colima and Jalisco, Mexico, reduced SWD populations by 50 to 55% (Gonzalez-Cabrera et al. 2019)”.
Regarding the interest for Argentinian fruit growers, the last paragraph of the discussion section refers to this issue (see next Response #5).
5.- Also, please explain why you chose non-crop hosts for your study and what that implies for cultivated hosts of the same plant species. In summary, since you combine ecology and management in your title, please do so in a more detailed way also in your discussion.
Response #5. Based on the reviewer 1's comment, the last paragraph of the discussion was modified as follows: “In conclusion, results reveal that both divergence in host choice and spatial patterns of same-resource preference among potential competitors, such as P. vindemiae and T. anastrephae, may mediate the coexistence of these two pupal parasitoids species in each natural environment tested in the current study. Given the apparent preference of the native T. anastrephae for resident saprophytic dipteran puparia mainly located in guava or peach fruit, P. vindemiae might be more suited to forage in less competitive microhabitats, such as the soil or outside of the fruit flesh, in which puparia of the exotic D. suzukii would naturally prevail in these habitats. From a SWD management approach,this scenario suggests that both pupal parasitoids have the potential as D. suzukii biological control agents. This is because such niche partitioning primarily involves differentiated exploitation of host microhabitats influencing the efficiency of both parasitoids in suppressing D. suzukii populations. Such an approach regarding the use of both pupal parasitoid species based on a differentiation in host microhabitat preference (soil vs foliage) was highlighted by Wolf et al. 2021 relying on semifield study results in Switzerland. Likewise, Kruitwagen et al. (2018) and Jarret et al. (2022) based on experimental adaptation studies of resident parasitoids to the invasive D. suzukii, pointed out that both T. anastrephae and P. vindemiae might offer greater potential to control SWD natural populations over larval parasitoids. Consequently, combining the two resident parasitoid species in wild non-crop environments may be an advisable alternative for local SWD management, either through augmentative releases (Rossi-Stacconi et al. 2019) or through a conservation biological control program (Lee et al. 2019). It is worth analyzing this initiative from an area-wide SWD management approach, as suggested by Rossi-Stacconi et al. 2019, Garcia [17], Wang et al. 2020, and Garcia et al. [11]. In this context, parasitoid releases should mainly be targeted to wild areas where known non-crop, alternative SWD hosts are abundant and may increase the risk of SWD infestations in surrounding fruit crops (Rossi-Stacconi et al. 2019, Wang et al. 2020, Wolf et al. 2020). Furthermore, pupal parasitoids would be more effective if released early in the fruiting season when SWD numbers are still low to avoid the pest population increase (Rossi-Stacconi et al. 2019, Wolf et al. 2020, 2021). This SWD biological control strategy is particularly relevant for the province of Tucumán, where feral guavas and peaches share the same geographical space with commercial berry orchards, as Tucumán hosts most of the soft fruit crops in fruit-growing regions of northern Argentina (Funes et al., 2018). Both feral fruit species allow the sustainability of SWD populations during the season in which commercial berry crops are not in production, representing a high economical risk for the local fruit industry. In this context, the use of both parasitoids is a practical and useful alternative for berry growers in Tucumán; they may release them in areas of wild vegetation adjacent to their crops or in orchards or backyards where there is no phytosanitary control. Finally, it is relevant to examine whether that niche differentiation in both parasitoid species occurs in other fruit host species, such as berries, and in other natural environments, such as berry crops or outlying areas to crop”.
Reviewer 2 Report
The article is well written. Methodology and results are descriptive and clear. The authors addressed one of the economically most relevant pest accross the globe thus any information on potential biological control agents and interactions of the fly with the parasitoids and their coexistence mechanisms are welcomed by the scientific community.
By my opinion the article should be accepted for publishing, but the authors are kindly asked to address two minor comments listed below.
Comment 1: Please, if possible, provide Fig 3 and Fig 6 with higher resolution
Comment 2: The authors have two abbreviations for Drosophila suzuki in the text i.e. SWD and Ds. There is no reason to have two. One of them (Ds) is used only in Abstract and not later in the text. Authors preferably used the second abbreviation SWS. Therefore, suggestion is to chose only one abbreviation and stick to it throughout the manuscript.
Also authors defined the abbreviations for parasitoids (Ta, Pv) and niches (IF, OF) but later throughout the text and on Figures didn’t use them. So, either give up on the abbreviations or have consistency in their use throughout the manuscript.
Author Response
Reviewer #2. Comments:
1.-The article is well written. Methodology and results are descriptive and clear. The authors addressed one of the economically most relevant pest accross the globe thus any information on potential biological control agents and interactions of the fly with the parasitoids and their coexistence mechanisms are welcomed by the scientific community. By my opinion the article should be accepted for publishing, but the authors are kindly asked to address two minor comments listed below.
Response. We appreciate the reviewer's very gratifying comments on our study.
2.-Comment 1: Please, if possible, provide Fig 3 and Fig 6 with higher resolution.
Response. We have improved figures 3 and 6 based on the reviewer's comments.
3.-Comment 2: The authors have two abbreviations for Drosophila suzuki in the text i.e. SWD and Ds. There is no reason to have two. One of them (Ds) is used only in Abstract and not later in the text. Authors preferably used the second abbreviation SWS. Therefore, suggestion is to choose only one abbreviation and stick to it throughout the manuscript. Also authors defined the abbreviations for parasitoids (Ta, Pv) and niches (IF, OF) but later throughout the text and on Figures didn’t use them. So, either give up on the abbreviations or have consistency in their use throughout the manuscript.
Response. Reviewer 2's comments are correct, we agree with his opinion, so we have rewritten the abstract based on his suggestions, as follows:
“Abstract: Understanding the mechanisms associated with the coexistence of competing parasitoid species is critical in approaching any biological control strategy against the globally invasive pest spotted-wing drosophila (=SWD), Drosophila suzukii (Matsumura). This study assessed the coexistence of two resident pupal parasitoids, Trichopria anastrephae Lima and Pachycrepoideus vindemiae Rondani, on SWD-infested fruit in disturbed wild vegetation areas of Tucumán, northwestern Argentina, based on niche segregation. Drosophilid puparia were collected between December/2016 and April/2017 from three different pupation microhabitats in fallen feral peach and guava. These microhabitats were “inside flesh (mesocarp)”, “outside flesh” but associated with the fruit, and “soil,” i.e., puparia buried close to fruit. Saprophytic drosophilid puparia (=SD) belonging to the Drosophila melanogaster group and SWD were found in all tested microhabitats. SD predominated in both inside and outside flesh, whereas SWD in soil. Both parasitoids attacked SWD puparia. However, T. anastrephae emerged mainly from SD puparia primarily at inside flesh, whereas P. vindemiae mostly foraged SWD puparia in less competitive microhabitats, such as the soil or outside flesh. Divergence in host choice and spatial patterns of same-resource preference between both parasitoids may mediate their coexistence in non-crop environments. Given this scenario, both parasitoids have the potential as SWD biocontrol agents”.